# Theoretical Study on the Aggregation and Adsorption Behaviors of Anticancer Drug Molecules on Graphene/Graphene Oxide Surface

**DOI:** 10.3390/molecules27196742

**Published:** 2022-10-10

**Authors:** Pengyu Gong, Yi Zhou, Hui Li, Jie Zhang, Yuying Wu, Peiru Zheng, Yanyan Jiang

**Affiliations:** Key Laboratory for Liquid-Solid Structural Evolution and Processing of Materials, Ministry of Education, Shandong University, Jinan 250061, China

**Keywords:** DFT calculations, MD simulations, adsorption and aggregation, graphene, graphene oxide, anticancer drugs

## Abstract

Graphene and its derivatives are frequently used in cancer therapy, and there has been widespread interest in improving the therapeutic efficiency of targeted drugs. In this paper, the geometrical structure and electronic effects of anastrozole(Anas), camptothecin(CPT), gefitinib (Gefi), and resveratrol (Res) on graphene and graphene oxide(GO) were investigated by density functional theory (DFT) calculations and molecular dynamics (MD) simulation. Meanwhile, we explored and compared the adsorption process between graphene/GO and four drug molecules, as well as the adsorption sites between carriers and payloads. In addition, we calculated the interaction forces between four drug molecules and graphene. We believe that this work will contribute to deepening the understanding of the loading behaviors of anticancer drugs onto nanomaterials and their interaction.

## 1. Introduction

Nanomaterials have broad application prospects in the biomedical field because of their unique characteristics. Drug delivery based on nanoparticles has been extensively studied to maximize the therapeutic efficacy of drugs [1]. Among the diverse nanomaterials that have been found, graphene and its derivatives have been demonstrated to provide efficient drug delivery and are considered as promising and ideal nanocarriers for drug delivery systems, and have been widely studied in the field of cancer treatment [2] due to their remarkable physical and chemical properties [2]. Graphene is a two-dimensional (2D) sheet of sp^2^ hybrid carbon atoms; the carbon atoms are tightly packed in a (2D) honeycomb lattice, which exhibits excellent properties such as large surface area and good biocompatibility, as well as providing a defect-free plane [3]. These pivotal characteristics allow it to interact with drugs through non-covalent interactions such as π-π interaction. As a derivative of graphene, GO is also a promising drug delivery vehicle [4]. Apart from features similar to pristine graphene, abundant hydroxyl, epoxy, and carboxyl functional groups in GO enable it to have a higher adsorption capacity for drug molecules than pristine graphene [5].

There are many drugs that could be delivered by graphene and its oxide, such as camptothecin, a widely used anticancer drug [6,7]. Its main target in cells is the type I DNA topoisomerase, which can inhibit DNA synthesis through chain break, causing cell death during the S phase of the cell cycle, making it an effective inhibitor of leukemia cell growth [8,9,10]. Based on this mechanism, Liu et al. studied the inhibitory effect of CPT on the growth of prostate cancer cells, as it can selectively inhibit the androgen-responsive growth of prostate cancer cells [11]. Therefore, CPT is also a potential candidate drug for the treatment of prostate cancer. Furthermore, resveratrol is a phytoalexin extracted in many edible plants that may play a role in preventing inflammation, atherosclerosis, cancer, and so forth [12,13]. For example, Kueck et al. found that Res inhibits glucose metabolism in human ovarian cancer cells. Zhou et al. [14] proposed that Res can induce apoptosis of pancreatic cancer cells. These studies suggest that Res is an effective cancer drug. In addition, gefitinib, an oral epidermal growth factor receptor (EGFR) tyrosine kinase inhibitor, is the first approved targeting drug for the treatment of non-small cell lung cancer (NSCLC) [15]. It has also been widely studied as a prospective drug for other cancers besides NSCLC, Li et al. [16] studied the potential role of Gefi in the treatment of pancreatic cancer and found that it can inhibit the growth, invasion, and colony formation of pancreatic cancer cells/Kalykaki et al. evaluated the effects of Gefi on circulating tumor cells (CTCs) in patients with metastatic breast cancer (MBC) [17,18]. Last but not least, anastrozole is a third generation aromatase inhibitor. As a potent inhibitor of intratumoral estrogen [19], clinical trials showed that Anas reduced the risk of breast cancer in postmenopausal women by 53% [20]. In the treatment of advanced breast cancer, it has significant survival benefits and tolerability advantages compared to other treatment drugs [21]. Therefore, it plays an important role in the prevention and treatment of breast cancer [22].

In this paper, the adsorption behavior of these drugs on graphene and GO carriers was investigated in depth using density functional theory (DFT) and molecular dynamics (MD) simulation, aiming to find the most stable adsorption conformations of different drug molecules on graphene and GO, and to compare the adsorption performance of the same carrier for different drugs. We hope that the results of this study can provide significant value for further design and development of new nanomaterial drug delivery systems, which we believe will ultimately improve the efficacy of targeted drugs in cancer therapy.

## 2. Computational Methods

### 2.1. Quantum Chemistry Calculations

We used quantum chemistry calculations methods to investigate the energetics of graphene and GO, and the effect of adsorption on drugs. DFT calculation is a quantum mechanical approach to study electronic systems and is commonly used to calculate the bind band structure of solids in physics. This method has been used for graphene-related research [23] All the quantum chemistry calculations were carried out with the Atomistix ToolKit (ATK) package. Generalized gradient approximation (GGA) [24] with Perdew–Burke–Ernzerhof (PBE) parametrization [25,26] was used as the exchange-correlation functional. The basis set consists of the double numerical atomic orbitals augmented by polarization functions, which are comparable to Gaussian 6–31G**. Compared with other methods, this calculation method is more effective and can meet the accuracy requirements [27,28]. To avoid neighboring interaction, the distance between the neighboring molecules was larger than 15 Å. The real-space global cutoff radii were set as 3.7 Å. The convergence criterion on the energy and electron density was set to be 10^−5^ hartree. Geometry optimizations were performed with convergence criteria of 2 × 10^−3^ hartree/Å on the gradient, and 5 × 10^−3^ Å on the displacement. 

The adsorption energy of CPT on to the studied nanosheets and GO is calculated using the relation: (1)$$\begin{array}{c} E_{a}=E_{complex}-E_{nanosheet}-E_{drug} \end{array}$$
where *E_complex_*, *E_carrier_*, and *E_drug_* are the total energy of the complex, energy of the carrier (GRA or GO), and energy of the drug molecule (Res, Ana, Gefi, or CPT).

### 2.2. Molecular Dynamics Simulation

MD simulation is a method of simulating molecules in chemistry using classical Newtonian mechanics with computer simulations [29] to obtaining material properties. MD has been widely used in the calculation of the materials such as graphene. Due to problems such as speed and difficulty in calculating large systems, we chose classical molecular dynamics as our research method. The force-field parameters were taken from the CHARMM force-field. We used the SwissParam web server to obtain the force-field parameters of the drug molecules. All the simulations were carried out by using the GROMACS 2018 software package. The initial structure of graphene containing 480 carbon atoms was constructed with the Nanotube Modeler package. To create GO, we randomly decorate the graphene surface with hydroxyl and epoxy groups. The final oxygen to carbon (O/C) ratio of GO nanosheets is 1:8. As for the relaxation of drug molecules, a box with a size of 4 nm × 4 nm × 4 nm was firstly established, small molecules were randomly inserted into the box, and the steepest descent method was used to optimize the system to remove close contact and overlapping. Since both sides of the graphene can be used for drug binding, it was placed in the middle of the box and the drug molecules were allowed to be randomly distributed on both sides. In all systems, the center of the graphene sheet was set as the zero point. Each system performed 10 ns NVT relaxation at 298 K and 1 atm, followed by 10 ns NPT-relaxation. After that, 50 ns MD simulation was conducted at 298 K and 1 atm equilibrium, and the integration step was 2.0 fs. The Berendsen thermal bath method was employed to control the temperatures. The cutoff radius of non-bonding interaction was set as 1.4 nm. Trajectories were collected every 5000 steps for further analysis. Visual molecular dynamics (VMD) was used to observe the movement trajectory of the system. 

## 3. Results and Discussion

### 3.1. Electrostatic Potential (ESP) of Drug Molecules

The reactivity and the interaction (especially for non-covalent interaction) of molecules can be determined by molecular surface electron density and electron activity, which is usually described by molecular electrostatic potential (ESP). In order to unveil the possible active sites in different drug molecules during drug adsorption, we have drawn electrostatic potential diagrams of different molecules, as shown in Figure 1. The red region represents positive electrostatic potential and shows electrophilicity, while the blue region represents negative electrostatic potential, which is more nucleophilic.

As shown in Figure 1a, graphene has uniform electron density and abundant π electrons on its surface. According to previous studies, graphene is prone to π-π electron donor acceptor interactions and van der Waals (vdW) interactions due to its large ring plane structure [30]. The graphene oxide shown in Figure 1b is plotted in blue at the oxygen atom. GO has reduced π electron activity to some extent due to the presence of oxygen-containing functional groups, but it may form hydrogen bonds with other molecules. The oxygen-containing functional groups of GO possess higher chemical reactivity compared to graphene. The ESP of Gefi is shown in Figure 1c, with lower ESP at the oxygen atom position. The ESP distribution of CPT is the same as that of Gefi, as shown in Figure 1d, with lower ESP near the functional group, which is more nucleophilic compared to the position of the hydrogen atom. The nitrogen atom position is shown in Figure 1e. The nitrogen atom position is plotted in blue with lower ESP, as shown in Figure 1f, and the oxygen atom position is plotted using red, indicating that the point has higher ESP [31].

### 3.2. Simulation of Graphene Adsorption of Drug Molecules on Graphene

As for the adsorption energy between graphene and drug molecules, we mainly focus on the parallel configuration of drug molecules due to the abundant π electrons on the graphene, which would easily result in the adsorption of drug molecules through vdW interactions. The optimized structures of graphene after adsorbing four drug molecules and their adsorption energy are shown in Figure 2. 

The vertical distance is the distance from the center of mass of the aromatic ring of the drug molecule to the plane of the graphene. The vertical distances are between the graphene and the aromatic rings of Gefi, CPT, Anas, Res are 3.358 Å, 3.462 Å, 4.991 Å, and 2.928 Å. The distance between graphene and different drug molecules follows the order: Anas > CPT > Gefi > Res; and their adsorption energy is Anas < CPT < Gefi < Res from small to large, indicating that the greater distance, the weaker the adsorption capacity of the drug is [32,33,34,35]. 

We used MD simulations to study the effect of drugs adsorption on the graphene and GO. We employed the root mean square deviation (RMSD), density distribution, radial distribution function (RDT), hydrogen bond number (H- bond) and mean square displacement (MSD) to investigate the dynamics process of the adsorption of drug molecules on graphene. 

Figure 3a shows the RMSD curves for different systems to investigate the equilibrium state of the simulated system. It can be seen that there are no large fluctuations in the RMSD curves of the system at the later stages of the simulation, indicating that it is sufficient to bring the system to equilibrium within the simulation time. As stated above, the main driving force for the adsorption of different drug molecules by graphene stems from the π-π interactions. The distribution of drug molecules on both sides of graphene was first investigated and the results are shown in Figure 3b. Their mass density shows that different drug molecules are effectively adsorbed on both side of the graphene sheet after the adsorption has reached equilibrium and that their distribution on both sides is symmetrical. Their two symmetry peaks are both at a distance of 0.38 nm, which is close to the vdW radius of the carbon atoms on the graphene sheet. In the range of distance less than 0.5 nm, all four kinds of drug molecules could appear on both sides of the graphene sheet. No drug molecule was observed beyond the range of 1 nm after the equilibrium of the system, because the mass densities of the drug molecules were all close to zero when the distances greater than 1 nm, which also indicates that their adsorption is relatively tight.

Radial distribution functions (RDF) can be used to study the intermolecular interaction. Figure 3c shows the interaction between drug molecules and graphene in the simulated system. There are significant interactions between different drug molecules and graphene. Their peaks are 0.482 nm for Gefi, 0.461 nm for CPT, 0.644 nm for Anas, and 0.436 nm for Res. There are only vdW interactions between the four drug molecules and graphene, and the strength of the interaction between drug molecules and graphene decreases as the distance increases. Res has the strongest interaction with graphene, while Anas has the weakest interaction. The adsorption capacities of graphene to the four drug molecules follow the order Res > CPT > Gefi > Anas. To further investigate the interaction between drug molecules and graphene, we calculated the probabilistic profiles of the distribution of the angle between the aromatic ring for drug molecules and the graphene plane in molecular dynamics. The ability of drug molecules to absorb on the graphene is mainly determined by the superposition of π-π interactions. Effective interactions between aromatic rings are considered to occur when the angle α < 30°. Figure 3d shows that all angles between the aromatic rings of four drug molecules and the graphene plane are small during the simulations. The most observed angles between Gefi, CPT, Anas, Res and the graphene were approximately 7°, 7°, 13°, and 8°, which indicates that their aromatic rings are almost parallel to the graphene surface. This also demonstrates that the adsorptions between drug molecules and graphene are stable and that π-π interactions are the main driving force for this adsorption. The environment will have a strong influence on the motion of the molecules in the system. Figure 3e shows the MSD results for different drug molecules adsorbing on graphene. Table 1 show the self-diffusion coefficients of the drug molecules in different systems. It can be seen that the diffusion coefficient of Anas is the largest, and the diffusion coefficients of CPT, Gefi, and Res are relatively similar, indicating that the diffusive motion of Anas is the most active among the four drugs. This result can be attributed to the weak binding ability of Anas and its low molecule weight (293.73), which facilitates its diffusion, whereas the other three drugs were strongly absorbed on the graphene surface and thus diffused more slowly [36,37].

### 3.3. Simulation of the Adsorption of Drug Molecules on GO

As for the adsorption behavior of drugs on GO, apart from the π-π interaction, drug molecules also generate hydrogen bonds with GO due to the functional groups, which promotes the adsorption of drug molecules on GO. According to Figure 1, we investigate the adsorption of different binding sites and named those structures as [GO—different drug names—X] (X = 1,2…). Several configurations were optimized and only two stable configurations for each kind of drug were selected to be further analyzed, which are illustrated in Figure 4. 

As shown in Figure 4a,b, the adsorption energies of stable configurations after Gefi adsorption on GO with active sites 2 and 4 of Gefi as binding sites are 1.823 eV and 1.553 eV, respectively, and the vertical distances are 3.387 Å and 3.451 Å between the aromatic ring of Gefi and GO, respectively. Thus, the adsorption of GO with Gefi is more stable when atom number 2 is used as the adsorption site, which is also consistent with the result of the electrostatic potential of molecule. Active sites 1 and 4 of the CPT were investigated as binding sites for adsorption onto GO, and their optimized result is shown as Figure 4c,d. The perpendicular distances between the aromatic ring of the CPT and the GO are 3.542 Å and 3.781 Å, respectively, and the adsorption energies are 1.326 eV and 1.037 eV, respectively; therefore, the configuration of the CPT molecules adsorbed on GO in Figure 4c can be considered as relatively stable. Figure 4e,f show the stable configurations obtained after adsorption on GO using active sites 1 and 2 of Anas as binding sites. The distances of the aromatic ring and GO are 6.416 Å and 3.431 Å, respectively. In Figure 4e, although the Anas molecule can form hydrogen bonds with GO, the vertical distance becomes larger, thus weakening the π-π interactions and leading to a reduction in the overall adsorption energy. Therefore, for Anas, the adsorption is more stable when binding at active site 2. As shown in Figure 1, all three active sites of Res have relatively large in electrostatic potential values and small differences; therefore, their adsorption with graphene is similar. Figure 4g,h show the result of the adsorption of active sites 1 and 2 of Res with GO. The configuration of the adsorption at active site 3 is not shown in the figure because it is consistent with that of active site 2. It can be seen that the vertical distance and adsorption energies of Res on GO are relatively close in both cases and, therefore, they are stable as binding sites for adsorption with GO in both cases. Overall, in the adsorption of four drug molecules onto GO, the final conformation is more stable when the binding site has higher electrostatic potential. According to the binding configuration and adsorption energy, the adsorption capacity of GO for the four drug molecules follows the order: Res > Gefi > CPT > Anas [38,39,40,41,42]. Comparing Figure 2 and Figure 4, the adsorption capacity of GO for four drug molecules is generally better than that of graphene. The vertical distance between different drug molecules and GO increased slightly, which weakened the π-π interaction between them to some extent, but the formation of hydrogen bonds promoted the binding of the two and the superposition of the two effects finally promoted the adsorption of drug molecules on GO. 

Figure 5a shows the RMSD result for the adsorption of four drug molecules on GO, from which it can be seen that all the systems reach the equilibrium state in a short time.

Previous studies have shown that not only π-π interactions, but also some hydrogen bonds occur during the adsorption of drug molecules on GO. To deeply understand the strength of the two kinds of interactions during the simulation, we investigated the average interaction energy between the four drug molecules and the GO sheet. As shown in Figure 5b. It can be seen that the vdW interaction accounts for the major part of the potential energy and is much greater than the Coulomb interaction, indicating that vdW interaction is the dominant force between GO and the drug molecules. This conclusion is also supported by the results of the radial distribution function of the simulated system [43]. 

Figure 5c shows the RDFs between different drug molecules and GO sheets. Four drug molecules had RDFs with GO in the range of less than 0.35 nm, indicating that the hydrogen bonding between them is relatively weak. Furthermore, the peaks at 0.486 nm, 0.612 nm, 0.662 nm, and 0.455 nm indicate vdW interactions between the four drug molecules and GO. This result is consistent with the analysis in Figure 5b, which indicates that vdW interactions between the four drug molecules and GO still play a dominant role, while the hydrogen bonding is relatively weak. 

The distribution of the angle between the aromatic ring of the drug molecule and the GO plane during kinetic adsorption was also analyzed; the results are shown in Figure 5d. The probability of the angular distribution between the aromatic rings of four drugs and the GO planes varied considerably compared to graphene during the simulation. Although the most probable angle between Gefi, CPT, Anas, and Res and GO remained relatively small, approximately 7°, 10°, 14°, and 9° respectively, the probability of occurrence decreased and, except for Res, there was a significant increase in the probability of the angle between the other three drug molecules and GO being greater than 30°. This also indicates that the π-π interactions between GO and drug molecules were weakened to some extent. This result mainly originates from the presence of oxygen-containing functional groups on GO. 

The hydrogen bonding in the system also plays an important role in the overall adsorption process; therefore, the variation in the number of hydrogen bonds between different drug molecules and GO during the simulation was investigated, as shown in Figure 5e. Throughout the adsorption process, it can be seen that the number of hydrogen bonds formed between all four drug molecules and GO is relatively small, which is also consistent with the results obtained in Figure 5b. This further confirms that the hydrogen bonding between the four drug molecules and GO is relatively weak during the adsorption process.

To further understand the movement of molecules in the system, the diffusion coefficients of the four drug molecules were studied separately, Figure 5f shows the MSD plots of drug molecules adsorbed on GO. Table 2 shows the diffusion coefficients of drug molecules in different systems in descending order. Comparing with Table 1, it is found that the diffusion coefficients of all four drug molecules show different degrees of reduction, which also indicates that the adsorption of these four drugs on GO is slightly stronger than that on graphene [39,44,45].

## 4. Conclusions

In summary, we used DFT methods and MD simulations to investigate the adsorption processes and interaction mechanisms of graphene and GO with Gefi, CPT, Anas, and Res. From the result of DFT calculations, it is clear that GO has stronger adsorption properties than graphene for the four drug molecules, and the adsorption energy follows the order of Anas < CPT < Gefi < Res for both the graphene and GO systems. Regarding the adsorption systems of drug molecules with graphene oxide, static calculations further confirmed the preferential adsorption sites. By utilizing MD simulations, we found the adsorption mechanisms of different drugs with graphene as well as GO; it was found that π-π interactions and hydrogen bonding played an important role in the whole adsorption process. Among the four drug molecules, Res molecules showed the strongest adsorption capacity on graphene and GO, while Anas showed the weakest adsorption capacity on both graphene and GO. Furthermore, vdW interactions played a dominant role in the dynamic adsorption of drug molecules on both graphene and GO. Hydrogen-bonding had only a small contribution to the adsorption of drug molecules on GO. Taken together, GO has a stronger ability to adsorb these four drug molecules than graphene. Due to the good adsorption properties of graphene and GO for the four drug molecules, this study helps gain insight into the loading behavior of anti-cancer drugs on graphene, and also helps to provide assistance in the development of carriers of loaded drugs for anti-cancer drugs.

## Figures and Tables

**Figure 1 molecules-27-06742-f001:**
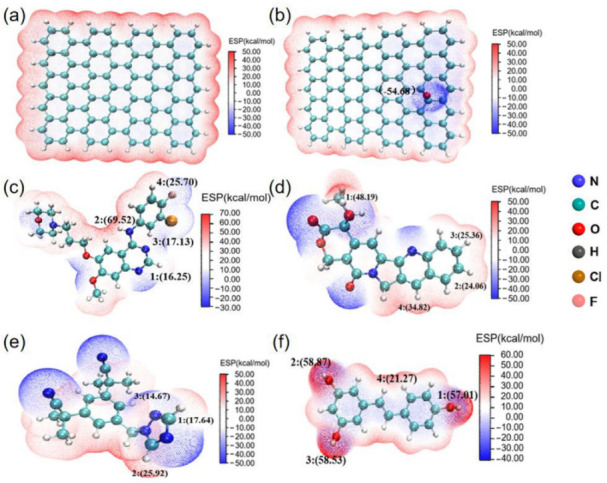
The electrostatic potential (ESP) distribution of (**a**) RGO; (**b**) GO; (**c**) Gefi; (**d**) CPT; (**e**) Anas; (**f**) Res.

**Figure 2 molecules-27-06742-f002:**
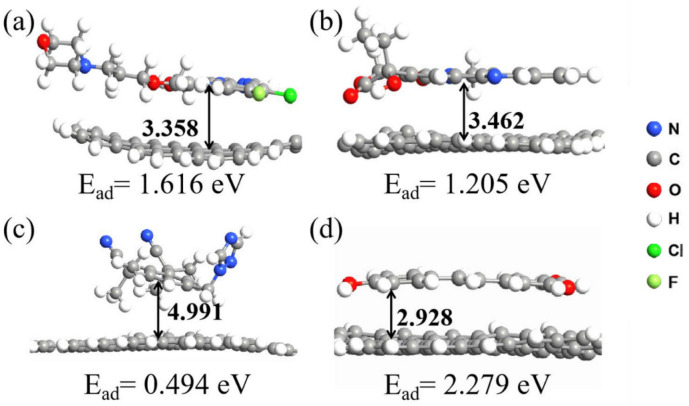
Optimized geometries of RGO and drug systems; bonds are in Å (the vertical distance refers to the distance between the centroid of benzene of different drugs to the carbon plane of RGO). (**a**) Graphene-Gefi; (**b**) Graphene-Camptothecin; (**c**) Graphene-Anas; (**d**) Graphene-Res.

**Figure 3 molecules-27-06742-f003:**
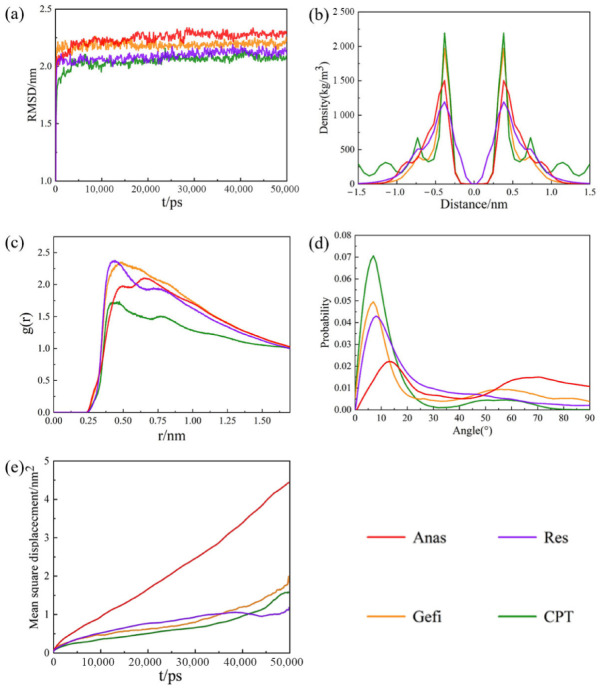
(**a**) The RMSD plots of the system with different drugs adsorbed on graphene as function of time. (**b**) Mass density profiles of different drugs. The center of graphene is set as distance = 0. (**c**) Radial distribution functions (RDF) of the different drug molecules with graphene. (**d**) The probability of the angle between the aromatic rings of the drug molecules and the graphene plane, (**e**) The MSD plots of the different drug molecules with graphene.

**Figure 4 molecules-27-06742-f004:**
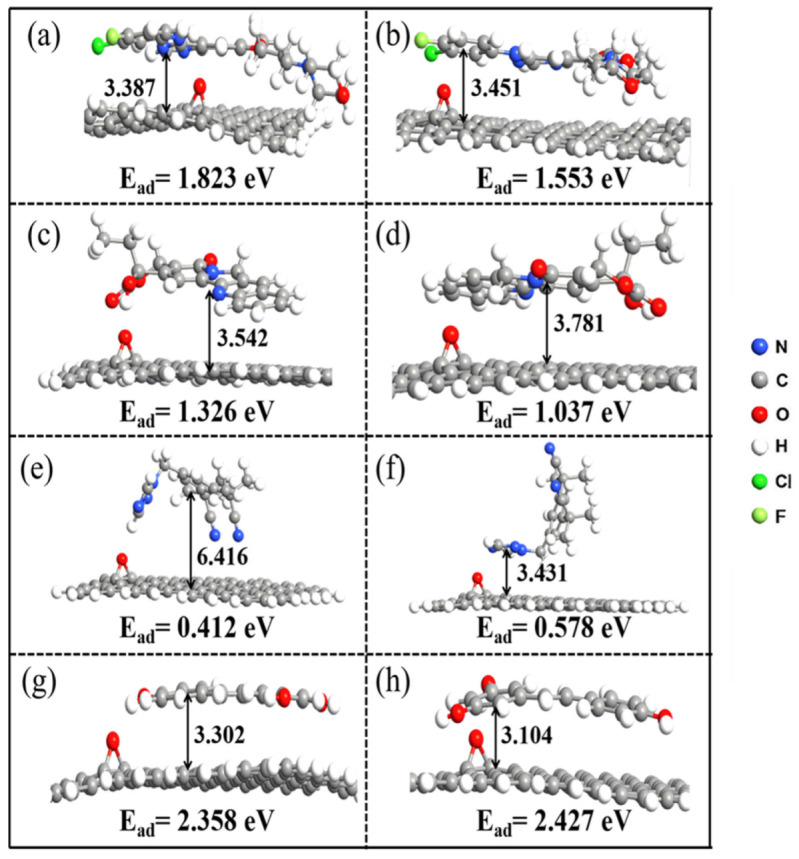
Optimized geometries of GO and drug systems; bonds are in Å (the vertical distance refers to the distance between the centroid of benzene of different drugs to the carbon plane of GO). (**a**) GO-Gefi-2; (**b**) GO-Gefi-4; (**c**) GO-Camptothecin-1; (**d**) GO-Camptothecin-4; (**e**) GO-Anas-1; (**f**) GO-Anas-2; (**g**) GO-Res-1; (**h**) GO-Res-2.

**Figure 5 molecules-27-06742-f005:**
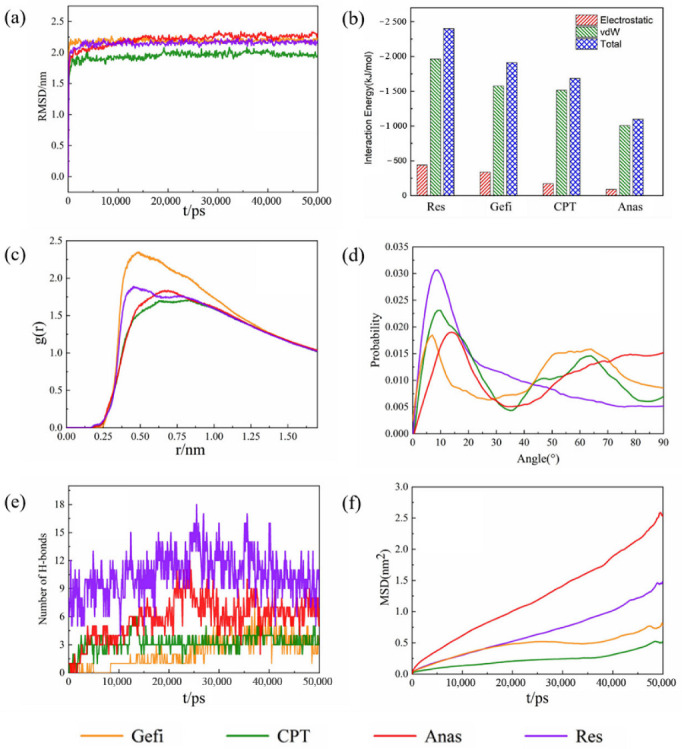
(**a**) The RMSD plots of the system with different drugs adsorbed on graphene as function of time (**b**) The average electrostatic, vdW, and total interaction energies of the systems with different drug molecules and GO. (**c**) Radial distribution functions (RDF) of the different drug molecules with GO. (**d**) The probability of the angle between the aromatic rings of the drug molecules and the GO plane. (**e**) Changes of the number of hydrogen bonds between the four kinds of drug molecules and GO over the time. (**f**) The MSD plots of the different drug molecules with GO.

**Table 1 molecules-27-06742-t001:** Self-diffusion coefficients of different drugs adsorbed on graphene surface.

Drug Molecules	Gefi	CPT	Anas	Res
Diffusion coefficient (×10^−5^ cm^2^·s^−1^)	0.004027	0.003376	0.0123	0.002793

**Table 2 molecules-27-06742-t002:** Self-diffusion coefficients of different drugs adsorbed on graphene oxide surface.

Drug Molecules	Gefi	CPT	Anas	Res
Diffusion coefficient (×10^−5^ cm^2^·s^−1^)	0.001416	0.001032	0.007013	0.002307

## Data Availability

Data is contained within article.

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
