# Peer review of "Theoretical Study on the Aggregation and Adsorption Behaviors of Anticancer Drug Molecules on Graphene/Graphene Oxide Surface"

_molecules, 2022, doi:10.3390/molecules27196742_

Round 1

Reviewer 1 Report

The manuscript describes a simulation study on the interaction of different anticancer drugs with graphene and graphene oxide in the context of biomedical applications.

In the abstract, it is mentioned that graphene is frequently used in cancer therapy. Has graphene been e.g. FDA approved? If you mean that graphene has been frequently studied to be potentially included in cancer therapies, then it should be clearly stated. No solvent models are clearly stated in the article, and yet the impact of solvent on the interaction will be essential for applications. The findings reported in the article will therefore be of minor importance to readers interested in biomedical applications that the manuscript is emphasizing. Moreover, the article focuses on the adsorption of molecules flat with respect to the substrate and does not consider the possibility of cluster formation or close-packing on the surface, which are likely to occur with such molecules, either upon adsorption, or when the system comes in the presence of water. Finally, the manuscript should be revised in depth to improve English grammar and spelling. For all these reasons I recommend rejection.

Reviewer 2 Report

This article is devoted to a theoretical study of the aggregation and adsorption behavior of molecules on the surface of graphene/graphene oxide. The studied molecules have anti-cancer properties, which undoubtedly adds to the relevance of this study. The article is written quite clearly and the calculations are not in doubt. There are the following remarks that need to be addressed:

1. Technical:

1.1 The design of the article must be strictly in the style of this journal. Use template.

1.2 It is desirable to increase the quality of drawings.

2. Content:

2.1 The abstract needs to be expanded.

2.2 When describing the choice of functional and method, it is necessary to describe why this particular functional and basis is best suited for this study.

2.3 Describe what other bases and methods are used for the theoretical study of these systems.

2.4 When describing figures and table values, it is desirable to add more references to the literature and compare the obtained data with data from the literature.

2.5 General - Please indicate how the theoretical data obtained can be applied in practice. What practical significance.

2.6 When describing the electrostatic potential, it is desirable to make a reference to the work: 10.1007/s00894-020-04645-5

Round 2

Reviewer 1 Report

Authors have a point that this is a model study in the absence of solvent. Looking forward to future comparison with experimental results and modelling in liquid.

Reviewer 2 Report

accepted